# Eye Movement Desensitization and Reprocessing versus Cognitive Behavior Therapy for Treating Post-Traumatic Stress Disorder: A Systematic Review and Meta-Analysis

**DOI:** 10.3390/ijerph192416836

**Published:** 2022-12-15

**Authors:** Ali Hudays, Robyn Gallagher, Ahmed Hazazi, Amal Arishi, Ghareeb Bahari

**Affiliations:** 1Community, Psychiatric and Mental Health Nursing Department, College of Nursing, King Saud University, Riyadh 11543, Saudi Arabia; 2Charles Perkins Centre, Faculty of Medicine and Health, The University of Sydney, Sydney 2050, Australia; 3Department of Public Health, Faculty of Health Science, Saudi Electronic University, Riyadh 13316, Saudi Arabia; 4Medical Surgical Department, College of Nursing, King Saud University, Riyadh 11543, Saudi Arabia; 5Nursing Administration and Education Department, College of Nursing, King Saud University, Riyadh 11543, Saudi Arabia

**Keywords:** post-traumatic stress disorder, eye movement desensitization and reprocessing, cognitive behavioral therapy, depression, anxiety

## Abstract

This meta-analysis review compared eye movement desensitization and reprocessing and cognitive behavior therapy efficacy in reducing post-traumatic stress disorder (PTSD), anxiety, and depression symptoms. A systematic search for articles published between 2010 and 2020 was conducted using five databases. The RevMan software version 5 was used. Out of 671 studies, 8 fulfilled the inclusion criteria and were included in this meta-analysis. Three studies reported that eye movement desensitization and reprocessing reduced depression symptoms better than cognitive behavior therapy in both children, adolescents, and adults (SDM (95% CI) = −2.43 (−3.93–−0.94), *p* = 0.001). In three other studies, eye movement desensitization and reprocessing were shown to reduce anxiety in children and adolescents better than cognitive behavior therapy (SDM (95% CI) = −3.99 (−5.47–−2.52), *p* < 0.001). In terms of reducing PTSD symptoms, eye movement desensitization and reprocessing and cognitive behavior therapy did not demonstrate any statistically significant differences (SDM (95% CI) = −0.14 (−0.48–0.21), *p* = 0.44). There was no statistically significant difference at the three-month follow-up and at the six-month follow-up for depression (*p* = 0.31), anxiety (*p* = 0.59), and PTSD (*p* = 0.55). We recommend randomized trials with larger samples and longer follow-up times in the future.

## 1. Introduction

Post-traumatic stress disorder (PTSD) is one of the most common causes of psychological disorders around the world. PTSD is defined as the exposure to several negative factors that affect people’s conditions and mood which could be threatened or experienced by a traumatic event [1]. The incidence rate of PTSD has been gradually increasing due to population growth and resulting conflicts and the prevalence and distribution of PTSD-related risk factors [2]. Individuals may be affected by PTSD in their lives due to natural and human-made events, including natural disasters, accidents, war, sexual assault, and serious illness [3].

Globally, PTSD affects nearly 4% of the population and is one of the most prevalent psychological conditions [4]. Few studies compared the incidence rates of traumatic events between different countries. The 3 countries with the highest rates are the Netherlands, Colombia, and the United States, whereas Italy, Romania, and Spain had the lowest exposure rates based on a comparison of 16 countries [5]. Moreover, different segments of the world population are at risk of developing PTSD. For example, female, individuals lacking social support, and the youth at exposure time have a high risk of developing PTSD [2]. A survey conducted in the United States on males and females aged between 16 and 76 years showed that more than 60% of the males experienced a traumatic event and more than 50% of the females were exposed to a traumatic event [6].

Patients with PTSD may suffer from other disorders’ symptoms. For example, comorbidity between PTSD and major depressive disorder (MDD) is common, and nearly half of PTSD patients also have a diagnosis of MDD [7]. In case PTSD and MDD co-occur, the same treatment plan provided for patients with PTSD could also help those with both disorders. PTSD patients could also have major distress disorder in response to a traumatic exposure [7]. Thus, a similar treatment strategy would be applied. Another important factor affecting post-traumatic symptoms’ adaption is self-efficacy. Post-traumatic symptoms may have a negative impact on self-efficacy, but self-efficacy may ease recovery from these symptoms over time [8].

Intentional and non-intentional traumas have been documented to have different impacts on the prevalence levels of PTSD. The prevalence of PTSD increased over time following an intentional traumatic event, whereas it decreased over time following a non-intentional traumatic event [9]. Further, the experience of trauma-related disorders during childhood not only influences one’s direct functioning but can have a long-term impact [10]. Traumatic symptoms resolve over time without any intervention in approximately 60% of individuals [9]. Thus, in case PTSD has evolved in those individuals, different psychological interventions such as self-help with support, counselling, psychoeducation, or selective serotonin reuptake inhibitors have been used [11]. However, the most recommended psychological interventions are eye movement desensitization and reprocessing (EMDR) and cognitive behavior therapy (CBT) [11].

EMDR treatment is one of the psychological treatments and has been debated about its efficacy for treating PTSD patients [12,13]. In the last decade, several trials have been conducted on this therapy and have been showing promising outcomes, though they are yet to be generalized. EMDR is a psychotherapy technique that is experimentally established for medical use to address the consequences of psychological trauma and other unpleasant life experiences. It enables patients to process distressing memories and replace them with new adaptive ones by focusing on past memories, present disturbances, and future actions [14]. EMDR can be performed by an expert therapist through using professional approaches, such as hand-tapping and audio simulation which aim to help patients to remember the worst moment of the trauma, and then treat it in subsequent sessions. This treatment is performed in eight phases [15], in which the patient continues to engage until the memory no longer distresses him or her. These phases are: (1) taking history and treatment planning, (2) preparing and explaining the treatment, (3) preparing the memory for the target, (4) memory desensitization, (5) teaching the patient a relevant positive belief, (6) identifying and processing residual disturbances related to the target, (7) ending the course session, and (8) re-assessing the patient to ensure he or she will be stable throughout the treatment and stay on track to meet his/her goals. Compared with pharmacological treatment, EMDR has much fewer side effects, is safer, and is widely regarded as a cost-effective and practical PTSD treatment [16,17].

CBT offers a fuller understanding of the processes that impact one’s emotional state by focusing on the cognitive individuals’ processes [18,19]. It is a psychological treatment that mainly use trauma-focused cognitive, behavioral, or cognitive-behavioral practices and exposure approaches to therapy [19]. The consequential advantage of this therapy is that results are visible quickly due to the effectiveness of the treatment. During extensive sessions, therapists do their best to understand the problem and adequately develop solutions. CBT also helps offer patients new insights into the world around them by helping them gain a deeper understanding of what they perceive around them. Such behavior therapy helps focus on the patient’s thoughts [19]. Recently, there have been many new forms of CBT developed for treating PTSD, including prolonged exposure (PE), brief eclectic psychotherapy, and cognitive restructuring [20]. The American Psychological Association strongly recommends PE in its professional practice guidelines for PTSD treatment. CBT has been proven as an effective treatment to reduce the negative thoughts associated with PTSD [21] because it addresses patients’ memories, thoughts, or feelings related to traumatic events [22]. Both EMDR and CBT include exposure and cognitive components. However, a main characteristic of EMDR is that patients are instructed to move their eyes quickly while concentrating on the distressing memory until the level of distress lessens. Patients also perform a relaxation exercise to help visualize a safe place and characteristics of the trauma and change negative ideas with positive thoughts. In CBT, the exposure intervention focuses on play or artistic expression instead of a formal written narrative [10].

Individuals with PTSD can also suffer from other psychological disorders such as anxiety and depression. Due to the increasing number of people who suffer from PTSD, effective treatments are desperately needed. Since no evidence is available about the efficacy of a specific treatment for both intentional and non-intentional traumas, EMDR and CBT remained the best treatment options for certain PTSD patients [11]. However, the efficacy of EMDR in PTSD treatment has been investigated in fewer numbers of trials [12]. Limited research also supported CBT to reduce the negative impacts of PTSD. Further, there is little research comparing EMDR with CBT for PTSD treatment. The effectiveness of both EMDR and CBT needs further confirmation to provide the best possible evidence. To our knowledge, no prior systematic reviews have focused on the significance of different PTSD symptoms’ treatments among individuals at different ages. Therefore, this meta-analysis review aimed to compare EMDR and CBT efficacy in reducing post-traumatic, anxiety, and depression symptoms among patients with PTSD. We also aimed to compare the cost-effectiveness and safety of both treatments to provide helpful knowledge on some factors related to both treatments.

### Main Question

The main question of this review is: How effective is EMDR compared to CBT in treating post-traumatic, anxiety, and depression symptoms among patients with PTSD?

## 2. Methods

### 2.1. Search Strategy

An electronic database search occurred using the following databases: Cumulative Index to Nursing and Allied Health Literature, PubMed, PsychINFO, Global Health via OvidSP (1910–present), and Scopus. The keywords included: “PTSD patients”, “traumatic events”, “post-traumatic stress symptoms”, “Eye movement desensitization reprocessing”, “Cognitive behavior therapy”, “treating PTSD symptoms”, and “reducing anxiety, stress, depression symptoms”. To combine the search terms or keywords, Boolean operators “AND” and “OR” were used. Incorporating the two operators and using different search terms led to a focused search that generated relevant journals according to the research question. A search limit was also applied to produce the appropriate studies through the application of filters. As a method of finding relevant, scholarly, and peer-reviewed studies, the filters included studies published between January 2010 and December 2020 for obtaining up-to-date evidence. Journal articles were selected by establishing eligibility criteria.

### 2.2. Selection of Studies

The inclusion criteria for the studies were that they should be: (a) focused on children, adolescents, and adults, (b) studies that stated a PTSD diagnosis, (c) studies that used reliable and valid measures, (d) randomized controlled trials (RCTs) comparing EMDR and CBT, (e) written in the English language, and (f) published between 2010 and 2020. Due to the capacity to identify causation and lack of bias, RCTs are the focus of this review. In the current review, only English publications were included, which can be construed as bias [21]. Nevertheless, most Gulf health publications which are related to mental illnesses and psychological therapies are published in English. A timeframe of ten years was judged sufficient to incorporate significant publications and recent research. Studies were excluded if their focus was on: (a) other treatments for PTSD symptoms, (b) descriptive articles, (c) books, (d) opinion articles, and (e) unrelated language.

### 2.3. Search Outcomes

Using the aforementioned databases, a total of 671 articles were identified, which were then selected based upon the Preferred Reporting Items for Systematic Reviews and Meta-Analyses (PRISMA) flow diagram [23]. Searching for grey literature and snowballing occurred to find any missing papers. As a first step, all relevant records were gathered from the five databases. After gathering all the records, the selection process began by eliminating duplicates from the databases using EndNote X9 software. As a result, 79 duplicate articles were identified. Afterward, non-relevant articles were removed after the initial review of titles and abstracts. There were 53 full articles remaining, but 45 were excluded because they did not meet the inclusion criteria. Ultimately, eight articles were included in this review and can be seen in the PRISMA diagram below (Figure 1).

### 2.4. Quality Assessment

A quality assessment determines with precision the value of evidence and whether it should be used in practice. Using critical appraisal tools provides researchers with the ability to apply their knowledge to practice. Systematic reviews are used to review and summarize the findings of the scientific evidence [23], although they cannot be assumed to be equally valuable regardless of how the study was conducted. In assessing the quality of the selected primary studies, the Cochrane risk of bias assessment tool was used [24]. Since this tool provides transparency to the evaluation of biases in internal and external validity, it was chosen for evaluating the eight RCTs. Due to the paucity of primary studies on this topic, none of the studies included in this review were excluded from the review based on a poor methodology, such as small sample sizes or insecure randomization.

### 2.5. Studies’ Appraisal

An appraisal has been performed for the included studies [3,25,26,27,28,29,30,31]. The aim was to evaluate the potential sources of bias of the studies included. The modified Cochrane tool measures the quality of different domains: selection, performance, detection, attrition, and reporting biases [24]. According to the severity of risk for each domain, bias will be assessed as ‘low risk’, ‘high risk’, or ‘unclear risk’. Only four studies gave an adequate description of how the randomization process was secured and carried out in terms of the randomization domain, which consists of two aspects: random sequence generation and allocation concealment [3,27,28,30]. The blinding of participants was not applied because of the kinds of interventions used, while the blinding of personnel was applied in four RCTs [25,27,30,31]. However, in all RCTs, the outcome assessors had been blinded to overcome the problem of detection bias.

Risk of attribution bias was not observed in any RCT due to adequate descriptions of how the missing data (e.g., patients dropped from the study) were treated in the outcome of the statistical analysis, which used intention to treat. Further, in the included studies, there were two strategies described to use intention to treat, descriptive statistics and the generalized linear model, which indicate a low risk of bias. Selective reporting bias was not found in any of the included RCTs. This is because all trials were registered on the ClinicalTrials.gov website before beginning to randomize participants. The second reason is that all study outcomes were reported. All potential bias risks are summarized (see Appendix A).

### 2.6. Data Analysis

We conducted meta-analyses assessing the differences of post-treatment and between three and six months of follow-up on post-traumatic, depression, and anxiety symptoms among patients with PTSD. Comparisons between included studies are presented in the results (see Table 1 and Table 2). We also calculated the weighted mean difference and 95% confidence interval of the included studies. The RevMan software version 5 was used to compare EMDR and CBT used in treating PTSD patients. Lastly, a review of the literature regarding the cost-effectiveness and safety of EMDR and CBT was conducted.

## 3. Results

### 3.1. Characteristics of the Included Studies

There were eight studies included in this systematic review and meta-analysis, and their characteristics are summarized in Table 1. The studies have been organized by date from oldest to most recent, as some studies were conducted by the same authors at different times. The eight included studies were published from 2011 to 2020 and evaluated the efficacy of EMDR and CBT for the treatment of patients with PTSD comorbid with other psychological disorders. Seven of them were from the Netherlands and one was from Australia. Out of the eight RCTs, three studies addressed the efficacy of EMDR treatment on children and adolescents (4–18 years old), and five studies were on adults (18–65 years old).

This review is comprised of eight RCTs published between 2011 and 2020, with a total of 780 participants including children, adolescents, and adults. The EMDR treatment group (n = 385, mean age ± SD: 28.19 ± 10) and the CBT treatment group (n = 395, mean age ± SD: 29.69 ± 10) were randomized to be included in the studies. It is important to note that in this review, studies that utilized mixed interventions with these treatment groups were excluded, as the study purpose was only to measure the efficacy of two treatment options with no other interventions.

In terms of the types of CBT variants covered in this study, there were a total of three RCTs that looked at the effectiveness of EMDR therapy and PE therapy on PTSD [25,30,31]. The other two RCTs compared brief eclectic psychotherapy with EMDR treatment for PTSD [28,29]. It is essential to highlight that Nijdam and colleagues conducted two RCTs, one published ten years ago and the second was published four years ago. The first study examined the effectiveness and response pattern of a trauma-focused CBT modality in treating PTSD, while the more recent study investigated between brief eclectic psychotherapy and EMDR for PTSD to determine how symptom improvements relate.

Further, another two RCTs were conducted by the same researchers in different timeframes [26,27]. The first study compared CBT to EMDR, while the second compared EMDR to a variant of CBT called cognitive behavioral writing therapy. In another study [3], the trauma-focused CBT modality was compared to EMDR to identify which one is more effective in treating PTSD. In terms of the study design, this review solely includes RCTs, five of which had two arms and three of which had three arms (see Table 2). Differences between measures for PTSD assessment are also provided in Table 2.

### 3.2. PTSD Symptoms Post-Treatment

Pooling seven studies in a meta-analysis there was little to no evidence to suggest that the difference in the mean level of post-traumatic symptoms from baseline to post-intervention differed for those who received EMDR treatment compared to those who received CBT (t6 = 0.77, *p* = 0.44). The difference is estimated at −0.14 fewer post-traumatic symptoms (SDM (95% CI −0.48 to 0.21)). In this quantitative meta-analysis, the included studies, however, had a high level of heterogeneity (I2 = 75%) (Appendix A). The included studies had no asymmetry or bias in a funnel plot of publication bias.

### 3.3. PTSD at Three-Month Follow-Up

The meta-analysis of two studies at a three-month follow-up indicates that there is little to no evidence that EMDR treatment reduces the post-traumatic symptoms in participants with PTSD compared to CBT (t1 = 0.60, *p* = 0.55). The difference is estimated at −1.09 fewer post-traumatic symptoms after the first treatment than the second therapy (SDM (95% CI −4.63 to 2.46)). However, the two studies had no level of heterogeneity (I2 = 0%) (see Appendix A). No asymmetry or bias in a funnel plot of publication bias was also found.

### 3.4. Depression Symptoms Post-Treatment

In this meta-analysis, depression symptoms were investigated through three studies. The meta-analysis showed that there is strong evidence that EMDR reduced the depression symptom levels compared to CBT (t2 = 3.20, *p* = 0.001). The difference is estimated at −2.43 lower depression symptom levels (SDM (95% CI −3.93 to −0.94)). However, the three studies had a low level of incidence of heterogeneity (I2 = 22%) (see Appendix A). Further, the included studies had no asymmetry or bias in a funnel plot of publication bias.

### 3.5. Depression Symptoms at Six-Month Follow-Up

At a six-month follow-up, the meta-analysis of one study showed that there was little to no evidence that EMDR reduced the depression symptom level compared to CBT (t = 0.54, *p* = 0.59). The difference is estimated at 5.10 lower depression symptom levels after CBT than after the EMDR (SDM (95% CI −13.29 to 23.49)) (see Appendix A). However, the heterogeneity was not applicable because only one study measured the follow-up at six months.

### 3.6. Anxiety Symptoms Post-Treatment

In this meta-analysis, 3 studies, which included 185 patients, revealed that there was very strong evidence that EMDR reduces the level of anxiety symptoms in patients with PTSD compared to CBT (t2 = 5.30, *p* < 0.001). The difference is estimated at −3.99 lower anxiety levels (SDM (95% CI −5.47 to −2.52)). However, the three studies had no level of heterogeneity (I2 = 0%) (see Appendix A).

### 3.7. Anxiety Symptoms at Three-Month Follow-Up

At a three-month follow-up, the meta-analysis of two studies showed that there was little to no evidence that EMDR reduced the anxiety symptom level compared to CBT (t1 = 1.01, *p* = 0.31). The difference is estimated at −3.49 lower anxiety symptom levels (SDM (95% CI −10.28 to 3.30)). However, the two studies had a low level of incidence of heterogeneity (I2 = 19%), as seen in Appendix A. In addition, the included studies had no asymmetry or bias in a funnel plot of publication bias

## 4. Discussion

This review was conducted to compare EMDR and CBT efficacy in reducing post-traumatic, anxiety, and depression symptoms among patients with PTSD. Both treatments were found to be effective at alleviating PTSD symptoms. In terms of alleviating depression and anxiety, EMDR treatment was found more effective. However, its effectiveness at reducing these symptoms was only available in post-treatment interventions and not evident at the follow-up.

### 4.1. PTSD Symptoms Post-Treatment

There are similarities and differences between the current meta-analysis findings and other meta-analysis findings that compared our two treatment options. The current meta-analysis results contradict a previous analysis [32], which did find a slight reduction but a statistically significant difference in PTSD symptoms with EMDR and CBT. The difference in results between studies might be explained by the inclusion of up-to-date papers in the current analysis and the exclusion of inappropriate papers that might have substantially contributed to the heterogeneity in the findings. In their meta-analysis, the heterogeneity was significantly evident, but the authors could not perform a further investigation due to insufficient studies. There is a statistically significant heterogeneity in our findings despite that the current meta-analysis involved two RCTs that had never been published in any review papers. The difference in results might also be attributed to the fact that the current review and the other meta-analysis had a small number of participants.

There was no statistically significant difference in terms of the efficacy of EMDR and CBT in the findings, as two studies showed that both treatments are effective in treating PTSD [33,34]. The difference found by Ho and Lee was similar to what was found in the current analysis. However, the incidence level of heterogeneity in the current meta-analysis was higher than Ho and Lee’s meta-analysis, which had low heterogeneity. The reason for the inconsistency is likely due to different measures being used for PTSD. For example, all studies in Ho and Lee’s analysis used identical measures for measuring PTSD, while the current analysis included studies that used different measures. It is also important to note that the studies included in Ho and Lee’s analysis were published up to 2006. On the other hand, the current analysis included studies published between 2011 and 2020, which is a major strength compared to the previous reviews. Moreover, EMDR treatment was determined comparable to CBT in reducing PTSD symptoms [35], which is consistent with the current meta-analysis.

### 4.2. Depression Symptoms Post-Treatment

Based upon the results of this meta-analysis, EMDR was significantly more effective than CBT at reducing depression symptoms post-intervention, which is incompatible with other prior studies, in which the authors reported no significant differences [35,36]. In contrast, in terms of depression reduction, other authors found a statistically significant difference between EMDR and CBT [32,34], which is consistent with the current analysis findings. There are some factors that play essential roles in the amelioration of depression, such as the length of the therapy session and the experience that therapists have in conducting group therapy [32].

In the current meta-analysis, we found that a range of 4 to 16 treatment sessions were used to treat post-traumatic, anxiety, and depression symptoms in patients with PTSD. Those in the EMDR group received from 15 to 90 min per session, whereas those in the CBT group received from 45 to 90 min per session and, thus, there is no consensus on how many sessions both modalities should use to treat these symptoms. According to some authors, between 12 and 20 sessions are required to see more lasting improvements in PTSD symptoms [37], although it might differ reliant on trauma type, the intensity of symptoms, and the client’s age. Psychological changes caused by trauma that took place over a longer period of time within adolescents and adults are likely to require more intensive EMDR treatment sessions [32]. This is similar to other authors’ recommendations that individuals with multiple traumatic events should obtain an adequate number of sessions, which range from 3 to 5 h per session, than those exposed to single event [37]. However, some studies included in the current review concluded that 6 to 12 sessions (15 to 90 min each) of EMDR were sufficient for improvement.

### 4.3. Anxiety Symptoms Post-Treatment

Anxiety disorder is more likely to be presented in PTSD patients due to negative experiences in the past, and they wish to avoid recalling those events. The current meta-analysis of three studies on the effects of EMDR compared to CBT on anxiety revealed that there is a significantly difference in favor of EMDR sessions. This result is consistent with analyses of another review [35], in which they reported the same result. It is important to highlight that our meta-analysis found a heterogeneity level similar to the heterogeneity level reported by Moreno-Alcazar et al. This shows that there is consistency among studies’ outcomes in the current and previous meta-analyses, which increases the confidence in recommendations about treatments for those diagnosed with PTSD anxiety symptoms. Similarly, other authors showed that there was a reduction in subjective distress in patients who had undertaken EMDR therapy in comparison to those awaiting treatment in a control group [38]. Therefore, this therapy helps patients cope with subsequent trauma by reducing anxiety and distress.

### 4.4. Cost-Effectiveness and Safety of Both Treatments

In light of the existence of the many treatments used for some mental disorders, it is necessary to highlight the characteristics of the selected treatment in terms of its efficacy, cost, and wanted benefits. It is becoming more common for policymakers, insurers, and providers to consider the effectiveness, costs, and benefits of different interventions when choosing among possible alternatives [39]. Generally, and especially concerning EMDR therapy, the costs and benefits of interventions have not been extensively studied. This therapy has been examined for its health and economic benefits in two studies. However, according to a research study conducted recently, individuals with PTSD and comorbid psychosis who received the EMDR treatment in addition to their standard therapy had more significant savings than those who received CBT for each patient per 6 months [40].

Another study evaluated the benefits and costs of PTSD treatment based on an analytical decision model provided by the British Mental Health Service [18]. Out of 11 different interventions evaluated in the study, the authors found that EMDR treatment was considered the utmost cost-effective approach to treating PTSD in adults. Contrary to their conclusion, CBT did not appear as cost-effective as other intervention modalities, such as psychoeducation and self-help with support. However, to generalize assessments, similar studies should be conducted in multiple different healthcare services.

EMDR treatment has remarkable safety characteristics, as it has far fewer adverse effects (such as light-headedness and vivid dreams) than pharmaceutical therapy [41] and is often well-tolerated by PTSD patients [18]. However, clinical practitioners may combine it with pharmacological therapies, such as sertraline and fluoxetine, in the most severe cases of generalized anxiety, depression, and schizophrenia. It appears that the synergy of pharmacological therapy combined with EMDR therapy is an effective and promising way to treat psychopathological disorders such as anxiety and depression [42]. Similar to treatment with CBT, PTSD from various traumas can be resolved using this type of therapy safely and effectively [43].

### 4.5. Strengths and Limitations

This meta-analysis had a number of strengths. First, the current analysis included up-to-date studies and the sample size of the included studies was larger than previous studies. Second, the studies included a wide range of age groups. The included studies also have good quality in design based on the Cochrane risk of bias tool, which might allow the results to be generalized. Further, the review did not include different countries and cultures. The non-diversity of countries may raise the generalizability of the findings. Some weaknesses were found in this meta-analysis. Some of the studies vary in heterogeneity, due to the lack of similarity in participants’ characteristics; thus, the validity of these results may be lowered. Moreover, in some studies there was a small sample size, which may restrict the ability to detect statistically significant differences in our meta-analysis.

### 4.6. Answering the Main Question

Despite the focus on EMDR and CBT interventions in each of the eight studies, there was also a comparison of different CBT approaches. This review included studies which focused on a number of treatment outcomes; for example, treating anxiety and depression alongside PTSD. Overall, EMDR treatment was found to be superior to CBT in treating anxiety in children and adolescents, but not in adults, and in treating depression in children, adolescents, and adults. In treating PTSD, however, no difference was found between the treatments. There are recommendations in most studies which emphasize that further research and larger sample sizes are needed for identifying the efficacy of both treatments in treating children and adolescents.

### 4.7. Implications

The level of awareness about PTSD has increased over the years, albeit not enough. However, to help increase the public’s understanding about PTSD and its therapy, it is suggested that some places, such as primary healthcare centers and public schools, could help spread awareness about this disorder by using brochures and posters. In addition, using reliable social media platforms might also be a powerful way of educating the general public. Educating patients’ parents or guardians also plays an important role because it helps them have a better understanding of the negative reactions that are caused by trauma, thus helping them better understand their children’s situation. As the child learns that these symptoms are related to PTSD, they will be reassured that their parents can provide the right treatment for them.

Based on the current findings, it can be assumed that the therapies delivered by qualified mental health professionals have been very effective in treating PTSD. EMDR and CBT are among these therapies, and both are shown to be effective in dealing with PTSD, according to the aforementioned results. Thus, researchers should use a bigger sample size and conduct more RCTs to examine the impact of them on PTSD. In terms of the management of anxiety and depression in PTSD, patients respond to EMDR better than CBT. This implies that researchers found evidence on the efficacy of the EMDR therapy for treating anxiety and depression in those patients, unlike in another study [35], which did not report any difference.

When it comes to EMDR therapy, some countries such as Saudi Arabia are still far behind since this kind of intervention is still considered relatively new. In Saudi Arabia, there is currently little to no study on using this therapy for treating psychological disorders, which indicates underutilization. However, this study might help encourage and support nursing staff and other healthcare professionals in evidence-based practice to learn more about EMDR therapy, thus helping to provide more effective care for PTSD patients in Saudi Arabia. In future studies, the effect of using such therapeutic intervention with a broader range of patients should be studied. Moreover, the studies in the future should pay more attention to the concerns of intervention costs.

## 5. Conclusions

The efficacy of EMDR therapy in reducing PTSD symptoms is still up for debate. Nevertheless, the research in this area has created huge interest between researchers themselves and physicians. In the included studies that were scrutinized for the purpose of the current study, it was found that the EMDR intervention has efficacy in treating children and adolescents with anxiety and is effective in treating children, adolescents, and adults with depression symptoms in people with PTSD. However, due to the lack of studies, it was not enough to give conclusive evidence about its efficacy. Moreover, the EMDR treatment was found to be as effective as CBT. It also has an advantage in time efficiency due to its results appearing in fewer sessions compared to CBT.

## Figures and Tables

**Figure 1 ijerph-19-16836-f001:**
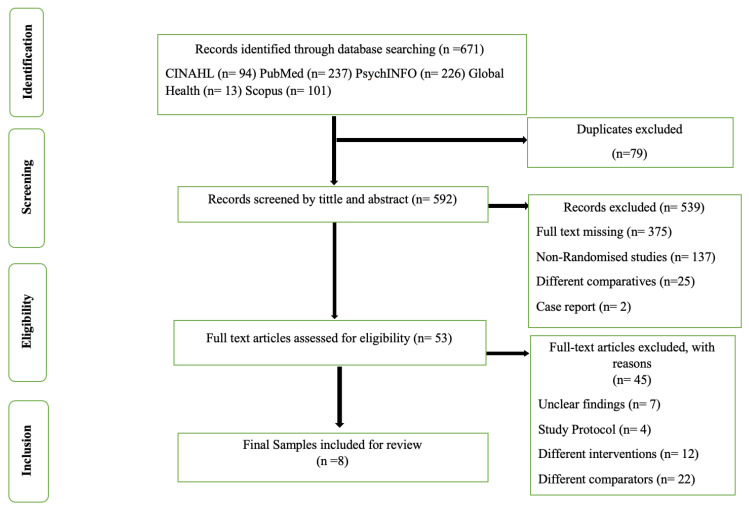
PRISMA flow diagram of the identification, screening, eligibility, and inclusion processes for studies.

**Table 1 ijerph-19-16836-t001:** Characteristics of the included RCT studies.

Citations	N	Country/Timeframe	(I)	(C)	Type of Trauma	Age, Mean (SD)		Males Females	Study Dropout	Follow-Up	Checking Treatment Fidelity	Psychological Comorbidity
de Roos et al. (2011) [26]	52	Netherlands 2001–2004	CBT N = 26	EMDR N = 26	Explosion of a fireworks factory (Single trauma)	CBT	EMDR	M: 29 F: 23	EMDR: 8 CBT: 6	3 months	Has been checked	Depression Anxiety Behavior problems
						10.01	10.2					
						(4.1)	(4.0)					
Nijdam et al. (2012) [28]	140	Netherlands 2003–2009	BEP N = 70	EMDR N = 70	Out-patients with PTSD (Mixed Trauma)	BEP	EMDR	M: 61 F: 79	EMDR: 20 BEP: 25	3 months	Checked	Anxiety Depression
						37.3	38.3					
						(10.6)	(12.2)					
Diehle et al. (2015) [3]	48	Netherlands 2009–2012	TF-CBT N = 25	EMDR N = 23	Single traumatic events	TF-CBT	EMDR	M: 18 F: 30	EMDR: 4 TF-CBT: 8	12 months	None	-
						9	9					
						(38)	(38)					
van den Berg et al. (2015) [31]	155	Netherlands 2011–2013	PE N = 53	EMDR + WL N = 55 + N = 47	Mixed trauma	PE	EMDR	M: 71 F:84	EMDR: 11 PE: 13	6 months	Checked	Psychosis
						42.6	40.4					
						(10.3)	(11.3)					
De Bont et al. (2016) [25]	155	Netherlands 2011–2013	PE N = 53	EMDR + WL N = 55 + N = 47	Traumatic psychotic experiences	PE	EMDR	M: 71 F: 84	EMDR: 12 PE: 8 WL: 7	6 months	None	-
						42.6	40.4					
						(10.3)	(11.3)					
de Roos et al. (2017) [27]	103	Netherlands 2010–2014	EMDR N = 43	CBWT + WL N = 42 + N = 18	Single trauma	EMDR	CBWT	M: 44 F: 59	EMDR: 1 CBWT: 1 WL: 2	12 months	Checked	Anxiety disorder
						13.41	12.96					
						(2.76)	(3.05)					
Nijdam et al. (2018) [29]	116	Netherlands 2003–2009	BEP N = 59	EMDR N = 57	Mixed trauma	BEP	EMDR	M: 55 F: 61	No	-	None	-
						37.56	39.53					
						(10.93)	(11.74)					
Stanbury, Drummond (2020) [30]	20	Australia 2011–2013	EMDR N = 10	PEN = 10	Traumatic memories	EMDR	PE	M: 6 F: 14	EMDR: 3 PE: 2	6 months	Checked	Depression Anxiety Stress
						40.1	45.5					
						(9.97)	(12.03)					

**Table 2 ijerph-19-16836-t002:** Characteristics and outcomes of the included studies.

Author Year	Sample Size Age Gender %	Intervention	Measure	Pre	Post	Second Post/3 Months/6 Months	Different Mean Change
**Three RCTs compare EMDR and CBT in Children and Adolescents**							
de Roos et al. (2011) [26]	52 10.2 years 55.7% male	EMDR using psychoeducation, then its protocol was applied on memory that has high levels of distress 1/eeek for 4–8 weeks/60 min per session. CBT using psychoeducation, focusing on exploration and correction on the behavior that is unwanted resulting from a traumatic memory through developing a trauma narrative. 1/week for 4–8 weeks	CROPS (PTSD) Interrater reliability (Cohen’s kappa: 0.96)	EMDR: 23.3 (9.9) CBT: 22.7 (9.6)	EMDR: 12.0 (9.1) CBT: 12.3 (8.1)	At 3 months: EMDR: 11.2 (8.0) CBT: 11.9 (8.3)	Pre to post: Change: 11.3 Change: 10.4 Effect size between groups 1.02–1.16 Pre to 3 months: Change: 12.1 Change: 10.8 Effect size between groups 0.98–1.10
			MASC (Anxiety) Cronbach’s alpha (0.88)	EMDR: 53.8 (17.7) CBT: 47.6 (16.8)	EMDR: 33.3 (17.4) CBT: 33.8 (18.4)	At 3 months: EMDR: 33.3 (17.4) CBT: 31.6 (18.4)	Pre to post: Change: 20.7 Change: 13.8 Effect size between groups 0.62–1.12 Pre to 3 months: Change: 20.5 Change: 16 Effect size between groups 0.85–1.02
Diehle et al. (2015) [3]	48 9 years 62.5% female	TF-CBT therapy was focused on children’s trauma, by gradual exposure by creating the child’s trauma narrative, and working on it by using psychoeducation, relaxation, affective expression and regulation, and cognitive coping. 1/week for 8 weeks (60 min per session). EMDR used desensitization of the memory for children with traumatic events through a weekly session for eight weeks (60 min per session).	CAPS-CA (PTSD) ICC for interrater reliability: 0.97–0.99	EMDR: 44.5 (19.4) TF-CBT: 42.3 (15.2)	EMDR: 23.6 (30.0) TF-CBT: 22.1 (23.3)	-	Change: 20.9 Change: 20.2 Effect size between groups 0.69 (95%CI 13.4, 14.8)
			RCADS (Anxiety) Cronbach’s alphas ranged from 0.75 to 0.95	EMDR: 4.1 (3.5) TF-CBT: 4.3 (3.7)	EMDR: 3.1 (3.7) TF-CBT: 3.1 (2.7)	-	Pre to post: Change: 1.2 Change: 1 Effect size between groups 0.24 (−3.0, 3.5)
de Roos et al. (2017) [27]	103 13.6 years 57% female	Parenting and children were involved in EMDR treatment sessions in the form of group discussions. Every week, parents share their observations about their child’s functioning 5 min before and after each session. 1/week for six weeks (45 min per session). Psychoeducation, promoting healthy coping strategies, cognitive restructuring, and writing narrative about a traumatic memory were included during CBWT sessions. 1/week for six weeks (45 min per session). After randomization, participants received recurrent appointments for six weeks, were advised they would be randomly assigned to EMDR or CBWT (if necessary), and the treatment began one week after allocation. In case of a crisis or much worsening of symptoms, WL participants were provided with a contact telephone number.	c-PTCI Child (PTSD) Cronbach’s alpha (0.78)	EMDR: 45.25 (13.12) CBWT: 48.44 (14.86) WL: 48.43 (15.69)	EMDR: 34.79 (12.34) CBWT: 36.56 (12.64) WL: 43.46 (14.09)	At 3 months: EMDR: 35.58 (14.07) CBWT: 37.36 (15.34) WL: No measure at follow-up.	Pre to post: Change: 10.46 Change: 11.88 Change: 4.97 Pre to 3 months: Change: 9.67 Change: 11.08 LMM: EMDR vs. WL (*p* = 0.03), CBWT vs. WL (*p* = 0.005), and EMDR vs. CBWT (*p* = 0.43).
			RCADS (Anxiety) Cronbach’s alphas ranged from 0.75 to 0.95	EMDR: 33.93 (19.88) CBWT: 43.89 (20.49) WL: 36.49 (20.83)	EMDR: 17.90 (19.18) CBWT: 24.63 (20.02) WL: 29.50 (18.09)	At 3 months: EMDR: 16.53 (17.55) CBWT: 22.88 (21.52) WL: No measure at follow-up.	Pre to post: Change: 16.03 Change: 19.26 Change: 6.99 Pre to 3 months: Change: 17.4 Change: 21.01 LMM: EMDR vs. WL (*p* = 0.01), CBWT vs. WL (*p* < 0.001), and EMDR vs. CBWT (*p* = 0.20).
**Five RCTs compare EMDR and CBT in Adults**							
Nijdam et al. (2012) [28]	140 38.3 years 56.4% female	BEP using psychoeducation, with an aim to relive the whole traumatic event in detailed imaginal exposure, writing assignments and cognitive restructuring were applied to participants. 1/week for 16 weeks (45–60 min per session). EMDR therapy: the distressing emotion was the target for examination and looking at that emotion in which part of the body, 1/week for 12 weeks (90 min per session).	HADS (Depression) Internal consistency reliability was assessed at 0.92	EMDR:10.93 (4.14) BEP:12.07 (4.05)	EMDR: 4.65 (4.39) BEP: 8.68 (5.57)	Second post: EMDR: 5.67 (4.54) BEP: 7.38 (6.42)	Pre to First post: change: 6.28 change: 3.39 Between groups *p* < 0.001 Pre to Second post: change: 5.26 change: 4.69 Between groups *p* = 0.13 Effect sizes from baseline to second post-assessment (Cohen’s d = 0.87 for brief eclectic psychotherapy and Cohen’s d = 1.21 for EMDR)
van den Berg et al. (2015) [31]	155 42.6 years 54% female	The PE therapy used imaginal exposure and vivo exposure based on a list of avoided trauma-related stimuli for participants. 8 weekly 90-min sessions within a 10-week timeframe. The EMDR therapy used the dual-attention stimulus for treating traumatic memories with participants. 1/week for 8 weeks (90 min) within a 10-week timeframe. The participants in the WL condition were seen once by a study therapist and informed about the PTSD diagnosis and further study course. After the 6-month follow-up period, appointments were made to begin their treatment of choice.	CAPS total score (PTSD) Consistency was measured at 0.81	EMDR: 72.1 (17.6) PE: 69.6 (14.9) WL: 68.1 (15.9)	EMDR: 40.3 (33.6–47.1) PE: 37.8 (31.2–44.3) WL: 56.5 (49.5–63.6)	At 6 months: EMDR: 38.8 PE: 36.7 WL: 51.9	Pre to post: change: 31.8 change: 31.8 change: 11.6 Effect size between groups 0.65–0.78 Pre to 6 months: change: 33.3 change: 32.9 change: 16.2 Effect size between groups 0.53–0.63
De Bont et al. (2016) [25]	155 42.6 years 54% female	The PE therapy used imaginal exposure and in vivo exposure based on a list of avoided trauma-related stimuli for participants. 1/week for 8 weeks (90 min). The EMDR therapy used the dual-attention stimulus for treating traumatic memories with participants. 1/week for 8 weeks (90 min). During the study, the therapist explained to participants once about the symptoms of PTSD and the further study course in the condition of WL. They then made appointments for their chosen treatment after the 6-month follow-up period	BDI-II (Depression) Reliability coefficient was 0.92	EMDR: 28.2 (55) PE: 30.9 (53) WL: 29.7 (47)	EMDR: 22.2 (44) PE: 18.3 (47) WL: 26.7 (39)	At 6 months: EMDR: 22.9 (43) PE: 17.8 (45) WL: 24.5 (39)	Pre to post: change: 6 change: 12.6 change: 3 Effect size between groups 0.42–0.78. Pre to 6 months: change: 5.3 change: 13.1 change: 5.2 Effect size between groups 0.15–0.64
Nijdam et al. (2018) [29]	116 39.5 years 52% female	BEP therapy: Imaginal exposure, cognitive restructuring and meaning making, and psychoeducation. 1/week for 16 weeks (45–60 min). EMDR therapy sessions for the remaining painful images, and the therapy is completed when the trauma memory has returned to normal status. 1/week for 16 weeks (90 min)	PTGI total (PTSD) Internal consistency reliability at 0.87	EMDR: 42.12 (17.93) BEP: 36.86 (19.62)	-	Second post: EMDR: 54.93 (23.77) BEP: 53.73 (22.14)	Pre to second post: change: −12.81 change: −16.87 Effect size between groups 0.05–0.65
Stanbury & Drummond (2020) [30]	20 45.5 years 70% female	The therapist led the participant’s eyes to follow the therapist’s fingers as they directed their attention to the memory, their negative belief, and the current body sensations during desensitization. 2/week for 6 weeks (15–90 min). Several sessions of PE therapy focused on the participant’s reactions to revisiting the trauma, integrating their thoughts, feelings, and meaning in life. 2/week for 6 weeks (90 min).	CAPS overall (PTSD) Coefficients between 0.90 and 0.97	EMDR: 86.71 (22.85) PE: 77.88 (13.07)	EMDR: 22.57 (21.68) PE: 17.13 (16.23)	At 3 months: EMDR: 21.43 (21.17) PE: 18.38 (13.26)	Pre to post: change: 64.14 change: 60.75 Between groups *p* = 0.09 Pre to 3 months: Change: 65.28 Change: 59.5

Abbreviations: EMDR: eye movement desensitization and reprocessing; CBT: cognitive behavior therapy; CROPS: Child Report of Post-Traumatic Symptoms; MASC: Multidimensional Anxiety Scale for Children; TF-CBT: trauma focused-cognitive behavior therapy; CAPS-CA: Clinician-Administered PTSD Scale for Children and Adolescents; RCADS: Revised Child Anxiety and Depression Scale; CBWT: cognitive behavior writing therapy; WL: wait list; c-PTCI: Children’s Post-Traumatic Cognitions Inventory; PE: prolonged exposure; CAPS: Clinician-Administered PTSD Scale; BEP: brief eclectic psychotherapy; PTGI: Post-Traumatic Growth Inventory; HADS: Hospital Anxiety and Depression Scale; BDI-II: Beck Depression Inventory; LMM: linear mixed model.

## Data Availability

The data presented in this study are available in the Appendix A.

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
