# Peer review of "Eye Movement Desensitization and Reprocessing versus Cognitive Behavior Therapy for Treating Post-Traumatic Stress Disorder: A Systematic Review and Meta-Analysis"

_ijerph, 2022, doi:10.3390/ijerph192416836_

Round 1

Reviewer 1 Report

I enjoyed reading this review on the treatment of post-traumatic, anxiety, and depression symptoms among patients with PTSD. While I generally think it is well-written and that the methods are sound there is a need for some editing, especially around the framing, and clarity of methods and results. In the literature there are many other reviews on the same topic that authors did not report (Khan A M, Dar S, Ahmed R, et al. (September 04, 2018) Cognitive Behavioral Therapy versus Eye Movement Desensitization and Reprocessing in Patients with Post-traumatic Stress Disorder:  Systematic Review and Meta-analysis of Randomized Clinical Trials. Cureus 10(9): e3250. doi:10.7759/cureus.3250) very similar to the present one. I have organized my comments by section.

Generally the writing is clear and well organized, however both the abstract and all paper need a careful edit to include acronyms of intervention approaches and reduce repetitions
- Cognitive Behavior Therapy (CBT), Eye Movement Desensitization and Reprocessing (EMDR), Prolonged Exposure (PE).

Introduction:
1. My largest concern is the framing of this study. Eye Movement Desensitization and Reprocessing (EMDR) is defined by authors as a therapy approach very different from Cognitive Behavior Therapy (CBT), but EMDR includes some components of exposure therapy but in the present review a theoretical foundation for understanding both neurobiologically informed trauma work and the need for a parts approach to treatment was not provided. Working with a parts approach allows therapists to work more effectively with complex and personality disorder clients.
More information need is about the theoretical constructs on which treatment approaches are founded.

On lines 70-71 the authors indicated the eight phases of a EMDR standardized protocol “3) preparing the memory for the target, 4) memory desensitization”. This is really a therapeutic element offered to participants derived from exposure therapy. Therefore, it seems like a very big stretch to make claims about the difference between EMDR and CBT. I would just be rethinking the description including more details and discussion.
2. I am confused about what this sentence “As opposed to psychotherapy, behavior therapy focuses on the patient's thoughts” reported on line 80 means. Authors should consider that behavior therapy is a psychotherapy. It might just need to be edited for clarity or better explained.

3. I am also confused about the sentence reported on line 269 “Both treatments were found to be superior at alleviating PTSD symptoms”. Superior in relieving PTSD symptoms than what?

Method
1. My largest concern here is that there measures from studies are not described. The item information, reliability and validity are all needed in characteristic of studies section. In the discussion (lines 292-294) it was mentioned that there was difference between studies regarding measures for PTSD assessment, but I failed to find any description.
Many more details are needed in this section.

2. Second largest concern is that there is information about the procedures that starts showing up in the discussion that should be in the methods.

3. It seems that the statistical analysis was run correctly.

4. I suggest enlarging Figure on PRISMA flow-chart.

5. I suggest enclosing a table with main studies characteristics and outcomes.

Discussion

Generally, there are things about the procedures that show up for the first time in this section that should move to the method, as cost-effectiveness. Lastly, I suggest a caution on EMDR may be most beneficial when there is a need for more immediate symptom reduction and at the early onset of trauma. While EMDR can be effective in the treatment of trauma, there are several other modalities that can be equally, if not more effective in treatment. These treatment options need to be discussed. Further research regarding co-morbidity and its impact on outcome as well as trauma and treatment history may be beneficial. Additionally, further research in longitudinal efficacy could be explored. It is, however, unlikely that any given therapy is universally appropriate for all individuals with PTSD and work is needed to develop more personalized approaches. We do not have a sufficient understanding of the efficacy of current therapies for those with a diagnosis of ICD11 complex PTSD. A potential future research implication could be type of trauma. Focusing on a specific trauma population such as childhood abuse or could narrow the results and provided more accurate direction of when treatment is most effective in practice. Further research in the areas of clinician experience and training may also be warranted along with different treatment variations.

I hope the authors will find my suggestions helpful to more clearly express your perspective preparing another version of the manuscript.

Author Response

Dear Ms. Pansy,

We have responded to all comments provided by the three reviewers (in one file). Please see the attachment

Thank you for your cooperation

Dr. Ghareeb Bahari

Author Response

Please see the attachment. All responses to the three reviewers' comments are included in the attached file

Reviewer 3 Report

The manuscript meets all the requirements for systemic review and meta-analysis articles. All the required elements are included and correctly described. In addition, the topic taken up by the authors is very interesting and increasingly common. The search for and development of reliable research findings on effective therapies for PTSD is valid and highly desirable. The reviewed manuscript is a valuable contribution to this field.

The manuscript contains minor editorial errors that can be quickly corrected, including that references to literature items should be corrected - they should be inserted in square brackets. The References section also needs to be adjusted to meet editorial requirements.

Author Response

Same thing...Please see the attachment. Our responses are included in the attached file

Round 2

Reviewer 1 Report

Thank you for the clear edits and responses. I think the manuscript is much stronger and now just needs some minor revisions. I appreciate the changes. Research questions are much clearer. On previous my review I invited authors to consider that:

On lines 70-71 the authors indicated the eight phases of a EMDR standardized protocol “3) preparing the memory for the target, 4) memory desensitization”. This is really a therapeutic element offered to participants derived from exposure therapy. Therefore, it seems like a very big stretch to make claims about the difference between EMDR and CBT. I would just be rethinking the description including more details and discussion. 

Concerning my frame: Further research regarding co-morbidity and its impact on outcome as well as trauma and treatment history may be beneficial. Additionally, further research in longitudinal efficacy could be explored. It is, however, unlikely that any given therapy is universally appropriate for all individuals with PTSD and work is needed to develop more personalized approaches. We do not have a sufficient understanding of the efficacy of current therapies for those with a diagnosis of ICD11 complex PTSD. A potential future research implication could be type of trauma. Focusing on a specific trauma population such as childhood abuse or could narrow the results and provided more accurate direction of when treatment is most effective in practice. Further research in the areas of clinician experience and training may also be warranted along with different treatment variations

For this point authors were requested to discuss complex PTSD also along the introduction, as the different interventions for specific trauma population. Sorry, but I not requested to the authors to copy my frame.
